# Integrated miRNA/mRNA Counter-Expression Analysis Highlights Oxidative Stress-Related Genes *CCR7* and *FOXO1* as Blood Markers of Coronary Arterial Disease

**DOI:** 10.3390/ijms21061943

**Published:** 2020-03-12

**Authors:** Miguel Hueso, Adrián Mallén, Ángela Casas, Jordi Guiteras, Fabrizio Sbraga, Arnau Blasco-Lucas, Núria Lloberas, Joan Torras, Josep M Cruzado, Estanislao Navarro

**Affiliations:** 1Department of Nephrology, Hospital Universitari Bellvitge, and Institutd’Investigació Biomèdica de Bellvitge-IDIBELL, 08908 L’Hospitalet de Llobregat, Spain; 15268jta@comb.cat (J.T.); jmcruzado@bellvitgehospital.cat (J.M.C.); 2Experimental Nephrology Lab, Institut d’Investigació Biomèdica de Bellvitge-IDIBELL, 08908 L’Hospitalet de Llobregat, Spain; addriann94@gmail.com (A.M.); angelita_2121@hotmail.com (Á.C.); jguiteras@idibell.cat (J.G.); nlloberas@ub.edu (N.L.); 3Department of Cardiac Surgery, Hospital Universitari Bellvitge, 08907 L’Hospitalet de Llobregat, Spain; sbraga@bellvitgehospital.cat (F.S.); arnaulasco@gmail.com (A.B.-L.); 4Independent Researcher; C/S. Albert Magne 12-14, Esc-B, SobAtic, Esplugues de llobregat, 08950 Barcelona, Spain

**Keywords:** atherosclerosis progression, counter-expression miRNA/mRNA analysis, oxidative stress, FOXO1, CCR7

## Abstract

Our interest in the mechanisms of atherosclerosis progression (ATHp) has led to the recent identification of 13 miRNAs and 1285 mRNAs whose expression was altered during ATHp. Here, we deepen the functional relationship among these 13 miRNAs and genes associated to oxidative stress, a crucial step in the onset and progression of vascular disease. We first compiled a list of genes associated to the response to oxidative stress (Oxstress genes) by performing a reverse Gene Ontology analysis (rGO, from the GO terms to the genes) with the GO terms GO0006979, GO1902882, GO1902883 and GO1902884, which included a total of 417 unique Oxstress genes. Next, we identified 108 putative targets of the 13 miRNAs among these unique Oxstress genes, which were validated by an integrated miRNA/mRNA counter-expression analysis with the 1285 mRNAs that yielded 14 genes, *Map2k1*, *Mapk1*, *Mapk9*, *Dapk1*, *Atp2a2*, *Gata4*, *Fos*, *Egfr*, *Foxo1*, *Ccr7*, *Vkorc1l1, Rnf7*, *Kcnh3*, and *Mgat3*. GO enrichment analysis and a protein–protein-interaction network analysis (PPI) identified most of the validated Oxstress transcripts as components of signaling pathways, highlighting a role for MAP signaling in ATHp. Lastly, expression of these Oxstress transcripts was measured in PBMCs from patients suffering severe coronary artery disease, a serious consequence of ATHp. This allowed the identification of *FOXO1* and *CCR7* as blood markers downregulated in CAD. These results are discussed in the context of the interaction of the Oxstress transcripts with the ATHp-associated miRNAs.

## 1. Introduction

Atherosclerosis (ATH) is a complex, multifactorial disease that affects physiological systems from the genomic to the cellular level. Oxidation of low-density lipoproteins (oxLDL)-cholesterol in the intima layer of vascular walls is the driver that provokes an inflammatory response characterized by the recruitment of monocytes to arterial walls in response to the secretion of chemokine C-C motif ligand 2 (*CCL2*) by endothelial cells [1].Further interaction of *CCL9* and *CCL21* with the chemokine C-C motif receptor 7 (*CCR7*) cause the retention of macrophages in inflamed vessel walls that contributes to chronic inflammation and plaque progression [2], while the accumulation of macrophage-derived foam cells exacerbates inflammatory signaling by producing reactive oxygen species (ROS) [3]. Oxidative stress (OS) is caused by the imbalance between the production of ROS and antioxidant defenses, and results in the net increase of reactive oxygen and nitrogen species (RONS). The vascular wall has a number of oxidant systems (e.g., xanthine oxydases, mitochondrial respiratory chain enzymes, NADPH oxidases, etc.) and antioxidant systems, such as superoxide dismutase (SOD), thioredoxins, and peroxiredoxins [4], whose expression is regulated by transcription factors of the forkhead box, class O (FOXO) family [5] and nuclear factor erythroid 2-like 2 (NRF2) [6] among others. Furthermore, a link between chronic kidney disease and cardiovascular diseases has been proposed that involves oxidative stress through mechanisms such as uremic toxin-induced eNOS uncoupling, increased NADPH oxidases activity, or antioxidant losses due to dietary restrictions, use of diuretic drugs [7], etc.

On the other hand, miRNAs are small post-transcriptional regulators involved in the control of mRNA function by establishing regulatory networks that can be altered in diseases (see [8] for a recent review). These regulatory networks can be very complex, since a single miRNA can interact with many different transcripts and these last can display binding sites for multiple miRNAs [9]. The identification of miRNAs and downstream mRNA-target interactions is an important topic to study regulatory miRNA/mRNA networks and their alterations in human disease. This has been approached with computer tools [10], that frequently originate a high number of false positive results [11,12], or with experimental techniques, that are complex and cumbersome to perform (see [13] and references therein). For these reasons, a number of groups have developed different protocols to integrate miRNA target predictions with expression data from microarray experiments in an attempt to facilitate the identification of miRNA/mRNA pairs without the difficulties of experimental models or the unacceptable high false positive rates of computational methods (see [14] and references therein).

Our group is interested in the role of miRNAs during atherosclerosis progression (ATHp). In a previous work, we used the ApoE^-/-^ murine model to profile changes in the expression of miRNAs and mRNAs upon systemic treatment with anti-CD40 specific siRNA, and identified 13 upregulated miRNAs and 1285 downregulated mRNAs [15,16]. Here, we have used a bioinformatics approach to deepen the understanding of miRNA-dependent regulation of gene expression in ATHp, with the result of identifying 14 transcripts likely involved in ATHp, and whose expression would be regulated by any of the 13 above upregulated miRNAs. These findings have been experimentally confirmed by profiling their expression in human samples of PBMCs obtained during coronary artery bypass grafting (CABG) in patients who suffered coronary arterial disease, a serious consequence of atherosclerosis progression. Whole blood gene expression profiling has the potential to be informative about underlying disease mechanisms and offers a genomic signature that allows to classify patients with cardiovascular diseases into finer categories [17]. This combined approach of bioinformatics and experimental validation has allowed the identification of *CCR7* and *FOXO1* transcripts as miRNA-regulated, Oxstress intermediates of coronary arterial disease.

## 2. Results

### 2.1. Compilation of the List of Genes Associated to the Response to Oxidative Stress (Oxstress Genes) by Reverse Gene Ontology Analysis (rGO)

We first compiled a list of oxstress genes by performing a reverse gene ontology search. GO is a widely used tool to convert long lists of genes into meaningful biological information through their classification in functional (or structural) groups termed “GO terms” [18,19]. Here, we have made the opposite operation, going from the GO terms to the genes, to extract lists of genes associated to GO terms. We first searched the gene ontology database with the primary GO term GO:0006979 (response to oxidative stress), which yielded a list of 400 genes that was loaded onto an Excel spreadsheet, and then we searched with the child terms GO:1902882, (regulation of response to oxidative stress, which yielded 94 genes), GO:1902883, (negative regulation of response to oxidative stress, which yielded 56 genes), and GO:1902884 (positive regulation of response to oxidative stress, which yielded other 23 genes). All these genes were loaded onto the same Excel column identified by their GeneSymbol unique identifier, and Excel functions were used to delete duplicates entries and to sort unique entries so that at the end of the process we had a list of 417 genes of response to oxidative stress that was saved for further analysis (Figure 1).

### 2.2. Integrated miRNA/mRNA Counter-Expression Analysis Identified Putative miRNA Targets Among Oxstress Genes

We next aimed to identify Oxstress genes that could be targeted by any of the 13 miRNAs upregulated in ATHp. We compiled a list of all predicted targets of these upregulated miRNAs by using the miRSystem browser (see Materials and Methods) wich yielded 3658 unique predicted targets, and used Excel functions to combine this list with the 417 Oxstress genes and to save entries common to both lists (108 transcripts). These represented Oxstress transcripts likely regulated by one or more of the 13 upregulated miRNAs in ATHp (Figure 1).

Common knowledge states that true miRNA targets should be counter-regulated with regard to their “parental” miRNAs, i.e., targets of upregulated miRNAs should be downregulated in the same experimental conditions. Thus, only Oxstress transcripts that were actually downregulated in ATHp could be “bona fide” targets of the 13 upregulated miRNAs. To identify these, we combined the list of 108 Oxstress transcripts with the 1,285 transcripts previously found to be downregulated by microarray hybridization during ATHp (Figure 1) and saved the 14 entries common to both lists. These were the following: ATPase Ca^2+^ transporting cardiac muscle slow twitch 2 (*Atp2a2*), chemokine C-C motif receptor 7 (*Ccr7*), death associated protein kinase 1 (*Dapk1*), epidermal growth factor receptor (*Egfr*), FBJ osteosarcoma oncogene (*Fos*), forkhead box O1 (*Foxo1*), Gata binding protein 4 (*Gata4*), potassium voltage-gated channel subfamily H (eag-related) member 3 (*Kcnh3*), mitogen-activated protein kinase kinase 1 (*Map2k1*), mitogen-activated protein kinase 1 (*Mapk1*), mitogen-activated protein kinase 9 (*Mapk9*), mannoside acetylglucosaminyltransferase 3 (*Mgat3*), ring finger protein 7 (*Rnf7*), and vitamin K epoxide reductase complex subunit 1 like 1 (*Vkorc1l1*), which were considered as validated Oxstress transcripts likely regulated by one or more of the 13 upregulated miRNAs in ATHp. Table 1 shows the identity of these 14 transcripts (as Seqnames and Refseqs) as well as data on their expression levels (fold change in absolute value) during ATH progression extracted from the previous mRNA profiling experiment. Table 1 also shows the output of other comparations that will be published elsewhere (see also Appendix A and Appendix A for the description of the experiments shown in Table 1 and the identification of the miRNAs used in each experiment). Finally, we also confirmed the relationship among miRNAs and the selected Oxstress transcripts by drawing a graphics of the predicted relationships among every Oxstress gene and each one of the 13 miRNAs. Interestingly, while we expected this comparison to give an “all against all” drawing, what we got was a kind of “X-shaped” diagram that suggested the existence of two main miRNA/mRNA networks of interactions (Figure 2 and Table 2). This could be due to the sharing of conserved miRNA binding sites among similar Oxstress transcripts, such as the MAPs. A more in deep description of these results will be published elsewhere.

### 2.3. GO Enrichment Analysis and Protein–Protein-Interaction Network Analysis (PPI) Highlight the 14 Validated Oxstress Transcripts as Components of Signaling Pathways

On a first glance, the list of 14 validated Oxstress transcripts showed a clear overrepresentation of genes linked to mitogen activated protein kinase (MAPK) signaling (Map2k1, Mapk1, Mapk9). To confirm this, we performed a GO enrichment analysis on the “GO Molecular Function” category. The output of the analysis is shown in Table 3 and in Appendix A, which show genes included in the top 20 functional categories recovered, most of them related to signaling pathways and to protein kinase binding and activity, such as transferase activity (*Map2k1*, *Egfr*, *Mapk9*, *Dapk1*, *Rnf7*, *Mapk1* and *Mgat3*) or MAP kinase kinase activity (*Map2k1*, *Mapk9*), but also to DNA-binding transcription factor activity (*Gata4*, *Foxo1*, *Fos*), transmembrane transporter activity (*Atp2a2*, *Kcnh3*), or signaling receptor activity (*Egfr*, *Ccr7*).

Subsequently, we performed a protein-protein-interaction network analysis (PPI) at the STRING database [20,21] (see Materials and Methods). STRING is a database of known and predicted protein-protein interactions that include direct (physical) and indirect (functional) associations. In the STRING output, individual proteins are seen as nodes and their interactions as edges that are color encoded according to their nature. Figure 3 shows the PPI analysis of the 14 validated Oxstress genes that was composed by 14 nodes (one for each one of the individual Oxstress proteins) and 32 edges linking them, a number of edges higher than that expected for random associations that indicated that these proteins were at least partially biologically connected. Furthermore, a more detailed analysis showed that the graphics followed a three-tier structure centered in the interactions among the components of a central cluster (*Map2k1*, *Mapk1*, *Mapk9*, *Fos*, *Egfr* and *Foxo1*), with their surrounding elements (*Dapk1*, *Atp2a2*, *Gata4*), and with satellite elements (*Vkorc1l1*, *Rnf7*, *Kcnh3*, *Mgat* and *Ccr7*), a result that highlights a role for MAP signaling in ATHp. 

### 2.4. Experimental Validation Highlights the Downregulation of CCR7 and FOXO1 in PBMCs from Human CAD Patients

Since all the above work was made with data obtained from model mice, we next aimed to confirm these results by measuring expression of a selection of the validated Oxstress transcripts by qPCR amplification of cDNAs obtained from samples of peripheral blood mononuclear cells (PBMCs) extracted from patients with coronary artery disease (CAD, *n* = 10) and from a control group without significant coronary lesions (nonCAD, *n* = 12, see patient’s demographics in Table 4). For this analysis we selected 9 genes, based on the output of the protein-protein-interaction network (Figure 3): six components of the interaction core (*FOS*, *MAP2K1*, *MAPK1*, *EGFR*, *FOXO1* and *MAPK9*), one surrounding node (*GATA4,*), two “satellites” (*CCR7* and *RNF7*), and *β-ACTIN* as internal control for normalization. Their expression was tested by duplicate in cDNAs from PBMCs from CAD patients by using TLDA cards, and the results were represented as a volcano plot of expression (as log_2_[Fold Change]) vs. statistical significance (as -log_10_[*p*-Value]). Figure 4A shows the result obtained in which only two of the Oxstress transcripts (*CCR7* and *FOXO1*) showed a significant downregulation. This result was backed by the scatter plot of the individual expression of *CCR7* and *FOXO1* in CAD and nonCAD samples (Figure 4B). *RNF7* was also downregulated but its *p*-Value did not reach statistical significance. On the contrary, *MAPK9* was upregulated but did not reach statistical significance either. The other two genes significantly downregulated in Figure 4A (*hnRNP-U* and *FTO*) belong to another experiment and will be published elsewhere. 

Lastly, we aimed to identify potential murine miRNAs targeting *Foxo1* and *Ccr7* to validate their human counterparts. We identified two of them (mmu-miR-30a-5p, mmu-miR-465a-5p) that were very promising since these targeted both, *Foxo1* and *Ccr7* (Table 2). Nevertheless, when we profiled expression of miR-30a in the human PBMCs, its expression was too low to be detected, and strikingly, mmu-miR-465a do not have a human homolog. 

## 3. Discussion

Our group is interested in the role of miRNAs in ATH progression. We previously tested the effect a gene therapy with a siRNA against the immune mediator CD40 (α-siCD40) on the development of ATH in which we profiled the expression of miRNAs and mRNAs and compiled a list of 13 upregulated miRNAs and 1285 downregulated mRNAs [15,16]. MicroRNAs (miRNAs) constitute a relevant tier of post-transcriptional regulation by their ability to form complex networks of miRNA /mRNAs, and disentangling these networks is critical to understand the mechanisms of gene expression. Nevertheless, there are severe technical drawbacks that pose limitations in this work, with direct experimental isolation of miRNA/mRNA hybrids being a complex and difficult task [22,23], while the use of bioinformatic tools rely in complex algorithms generates many false positives and produces lists with hundreds of putative targets, challenging further downstream processing (see [24,25], and [26] for a recent review). Here, we have presented a combined experimental-computational approach to search for miRNA/mRNA pairs related to the response to oxidative stress (oxstress) in PBMCs from human patients with coronary artery disease (CAD), a complication of atherosclerosis progression, with the ultimate aim of detecting markers of CAD progression in blood. To do this, we have integrated data from public databases with our previous data on miRNA and mRNA expression in ATHp, and with the data herein generated on the expression of a selected group of Oxstress genes in PBMCs from patients with severe coronary artery disease (Table 4).

We first compiled a list of Oxstress genes by performing a reverse Gene Ontology search in which we extracted all genes from the GO term “response to oxidative stress” and its child terms (identification step, see Materials and Methods). We next selected those that also appeared in the list of 3658 predicted targets of any of the 13 murine miRNAs upregulated in ATHp from our previous work with the ApoE-deficient mice [15,16] and shortened this list to those that were actually downregulated during ATHp in the animal model (validation step). To identify these, we combined the list of 108 Oxstress transcripts with the 1285 transcripts previously found to be downregulated by microarray hybridization during ATHp (Figure 1) and saved the 14 entries common to both lists (see Materials and Methods). These were the following: *Atp2a2*, *Ccr7*, *Dapk1*, *Egfr*, *Fos*, *Foxo1*, *Gata4*, *Kcnh3*, *Map2k1*, *Mapk1*, *Mapk9*, *Mgat3*, *Rnf7* and *Vkorc1l1*, which were considered as validated Oxstress transcripts likely regulated by one or more of the 13 upregulated miRNAs in ATHp. Some of these 14 genes have been directly related to ATHp. Thus, *Atp2a2* was shown to regulate the NO-induced contractile phenotype of VSMCs [27] while its activity was repressed by oxidation by NADPH oxidases [28], *Gata4* activated SMC proliferation by binding to the cyclin D1 promoter [29] and regulated positively *Nox4* expression [30], inhibition of *Egfr* reduced T cell infiltration in ATH and protected against ATHp [31], fibrosis-related *FOS* and *JUN* transcription factors were up-regulated in foam-cell macrophages [32], while inhibition of *Map2k1* was shown to be anti-atherogenic [33], the knockdown of *Mapk1* was shown to suppress the development of coronary atherosclerotic heart disease [34], etc. 

Lastly, to confirm the above results, we measured the expression of a selection of the validated Oxstress transcripts in samples of PBMCs extracted from patients with coronary artery disease (CAD, *n* = 10) and a control group without significant coronary lesions (nonCAD, *n* = 12, see patients demographics in Table 4). Oxidative stress and the generation of free radicals in the vessel wall are crucial in the onset and progression of vascular disease, and the severity of CAD has been associated with elevated plasma levels of end products of lipid peroxidation, a marker of oxidative stress and a reduced antioxidant capacity [35]. For this analysis we selected nine genes, based on the output of the protein–protein-interaction network (Figure 3): four components of the interaction core (*FOS*, *MAP2K1*, *MAPK1* and *MAPK9*), three surrounding nodes (*GATA4*, *EGFR* and *FOXO1*), and two “satellites” (*CCR7* and *RNF7*) and we could identify that only two of the Oxstress transcripts (*CCR7* and *FOXO1*) showed a significant downregulation (Figure 2). The identification of *CCR7* as intermediate of CAD development is not surprising since this chemokine receptor together with its ligands *CCL19* and *CCL21* conforms an axis that has been characterized as a regulator of the maturation and migration of T-lymphocytes and lymphoid tumor cells (see [36,37] for reviews), and CAD has a deep inflammatory component [38]. *CCR7* has been shown to have a positive role in plaque regression [2,39], likely by facilitating the emigration of macrophages from plaques [40,41], while its genetic depletion was seen to increase T cell accumulation in atherosclerotic lesions and to aggravate ATHp [42], results in agreement with our here reported downregulation of *CCR7* expression in CAD patients. Interestingly, CCR7 was shown to protect from the effects of oxidative stress [43] while CCR7^+^ T-cells probed to be sensitive to the effects of H_2_O_2_-induced oxidative stress [44].

Another very interesting finding of this work is the detection of a downregulated *FOXO1* expression in PBMCs of CAD patients. *FOXO1* is a member of the forkhead-box O family of transcription factors that have key roles in the maintenance of tissue homeostasis (see [45] for a recent review), in the cardiac regulation of glucose and lipid metabolic pathways [46], in the development of the adaptive immune response by promoting the formation of T-cells and the maturation of B-cells and protecting them from oxidative stress [47], in the regulation of adipocyte differentiation [48] etc. On the other hand, *FOXO1*, whose expression is sensitive to reactive oxygen species, has an important role in cellular oxidative stress resistance pathways by activating transcription of antioxidant genes [5,49]. *FOXOs* have been also linked to the development of ATHp and CAD by inhibiting proliferation and activation of VSMCs and neointimal hyperplasia [50], by mediating protease activated receptor 2 (PAR2)-induced proinflammatory gene expression [51], etc. Lastly, *CCR7* and *FOXO1* are functionally linked since *CCR7* expression is regulated by FOXO1 [52] and both are predicted targets of miR-30a-5p and miR-465a-5p, two of the 13 miRNAs upregulated in ATHp (Table 2). Nevertheless, when we aimed to measure the expression of miR-30a-5p and miR-465a-5p, we found that mmu-miR-465a-5p does not have a human homolog, something not so surprising since, in mice, the overexpression of this microRNA has been only associated to aging [53]. On the other hand, the lack of detection of miR-30a in human PBMCs could indicate that its expression is tissue-restricted. In this sense, it has been reported that expression of miR-30a was increased in aortic dissection samples [54], and that downregulation of the miR-30 family contributed to the Endoplasmic reticulum stress in the cardiovascular system [55]. Although the functional relationship among *CCR7*, *FOXO1*, miR-30a, and miR-465a could not be proved, there are other miRNAs that could be potential targets of these transcripts (Table 2 and Figure 2), and work is in progress to test these associations. 

## 4. Materials and Methods

### 4.1. Reagents

In this work, we used the following reagents Ficoll (SepMATE^TM^ PBMC Isolation Tubes, StemCell technologies, Vancouver, BC, Canada), Maxwell RSC miRNA Tissue Kit (Cat.# AS1460, Promega, Madison, WI, USA), TaqMan^®^ Advanced miRNA Assays–single tube assays (Cat.#A25576, Thermo Fisher, Waltham, MA, USA), TaqMan^®^ Advanced miRNA cDNA Synthesis Kit (Cat.# A28007, Thermo Fisher, Waltham, MA, USA). Taqman Gene Expression Assay (Thermo Fisher, Walthman, MA, USA) were used to measure expression of hsa-miR30a-5p (Assay ID: 000417) and has-miR16-3p (Assay ID:002171).

### 4.2. Murine miRNA and mRNA Expression Data

ATHp experiment in ApoE-deficient mice was approved by the ethic committee for animal research of UB-Bellvitge and published elsewhere [16]. Briefly, we used aortic total RNA from ApoE-deficient mice at basal conditions (8 weeks old), from mice treated with an α-CD40 siRNA (SiCD40) for 16 weeks, or from mice treated with a scrambled control siRNA (siSC) also for 16 weeks. MiRNA expression was analyzed by using TaqMan Low Density Array cards (TLDAs) and mRNA expression was tested by microarray hybridization on a commercial basis [15,16].

### 4.3. Patients

We used stored material from 22 patients undergoing cardiac surgery at Bellvitge University Hospital. Among them there were 10 patients with Coronary artery disease (CAD) collected from coronary artery bypass grafting (CABG) patients and 12 controls from surgical valvular replacement patients without coronary disease (nonCAD). Subjects receiving any antidiabetic medication were considered diabetic. The study was authorized by the local ethics committee of Bellvitge University Hospital (PR173/18, approved on 5 September 2018) and all patients provided written consent.

### 4.4. Human Samples and RNA Extraction

Blood samples were obtained during the anesthesia induction. PBMC were isolated less than 2 h after blood collection using ficoll (SepMATE^TM^ PBMC Isolation Tubes, StemCell technologies, Vancouver, British Columbia, Canada) and frozen in a DMSO-containing solution. We used frozen cells for the transcriptomic analysis. Total RNA was extracted from PBMC by using the Maxwell RSC miRNA Tissue KIT (Promega, Madison, WI, USA).

### 4.5. Expression Profiling of Oxstress Transcripts in Custom-Made TLDA Cards

cDNA templates were synthesized from PBMC total RNA with the TaqMan^®^ Advanced miRNA cDNA Synthesis Kit (Cat# A28007, Thermofisher, Waltham, MA, USA). All real-time quantitative PCRs were performed by duplicate in a 384-well format (TaqMan^®^ Low Density Array Cards, TLDAs) on a 7900HT Fast Real-Time PCR Instrument (Thermofisher, Waltham, MA, USA) and using TaqMan^®^ Fast Advanced Master Mix. The thermal protocol was adjusted for the TaqMan^®^ Fast AdvancedMaster Mix. IDs for individual Oxstress transcript assays in the cards were as follows: *FOXO1* (Hs00231106_m1), *MAP2K1* (Hs05512159_s1), *MAPK1* (Hs01046830_m1), *MAPK9* (Hs01558224_m1), *FOS* (Hs04194186_s1), *EGFR* (Hs01076090_m1), *CCR7* (Hs01013469_m1), *GATA4* (Hs00171403_m1), *RNF7* (Hs02621493_s1), *GAPDH* (Hs99999905_m1), and *β-ACTIN* (Hs01060665_g1). Hsa_miR-30a-5p expression (Assay ID: 000417) were normalized using has-miR-16-3p (Assay ID:002171) as endogenous reference gene. Expression analysis was performed using the comparative CT (ΔΔCT) method with the Expression Suite software v1.1 (Thermofisher, Waltham, MA, USA) using β-actin as endogenous control for normalization.

### 4.6. Reverse Gene Ontology (rGO) Identification of Genes Involved in the Response to Oxidative Stress

Oxstress genes were identified by searching the “Gene Ontology” browser (http://geneontology.org) [18,19] with the GO terms GO:0006979 (response to oxidative stress), GO:1902882 (regulation of response to oxidative stress), GO:1902883 (negative regulation of response to oxidative stress) and GO:1902884 (positive regulation of response to oxidative stress). The filter for “murine genes” was activated and the link “display genes and gene products annotated to response to oxidative stress” was followed. All the entries were copy-pasted onto an Excel spreadsheet, pruned of all associated data except the “Genesymbol” unique identifier, sorted and examined to eliminate duplicated entries by using the “sorting” and “Data > Remove duplicates” functions of Excel. This generated a list of 417 genes linked to Oxstress in Excel format.

### 4.7. Detection of Predicted miRNA/mRNA Targets

The list of 13 miRNAs upregulated during ATH progression in the 24 weeks old mice group treated with a scrambled SiRNA for 16 weeks (SC24W) was loaded onto the miRSystem browser (http://mirsystem.cgm.ntu.edu.tw), that uses seven different popular algorithms to predict miRNA targets [11], and the search was run with the default settings. All the potential target genes were copy-pasted onto an Excel spreadsheet, duplicated entries were eliminated as above, and the list was cleaned of all information but the official “GeneSymbol” and the “gene name”. This generated a list of 3658 predicted miRNA unique targets in Excel format (Hueso et al., 2019, In preparation). To identify miRNAs targeting the validated Oxstress genes, these were used to interrogate individually the miRSystem browser. 

### 4.8. Counter-expression miRNA/Target mRNA Analysis

The Excel list of 3658 predicted miRNA unique targets and the list of 417 genes involved in the response to oxidative stress were loaded onto the same column of a new Excel sheet. For the counter-expression analysis, Excel functions were used to alphabetically sort both lists together and to detect and extract entries common to both lists which corresponded to Oxstress genes potentially targeted (downregulated) by any of the 13 upregulated miRNAs. These were validated by comparing them with the list of 1285 mRNAs which were experimentally seen to be downregulated at SC24W vs B8W [16]. All these analyses were made by using the “GeneSymbol” as unique identifier for all genes/transcripts involved. 

### 4.9. GO Enrichment Analysis

The Gene Ontology (GO) analysis [18,19] was performed in the “ShinyGO v0.60: Gene Ontology Enrichment Analysis” browser at http://bioinformatics.sdstate.edu/go/ [56]. “Mouse genes”, “*p*-value (FDR) = 0.05” and the “GO Molecular Function” Functional Category were selected for the analysis.

### 4.10. STRING Protein-Protein Interaction (PPI) Network Analysis

PPI network analysis was performed at the STRING database (https://string-db.org) [20,21] after loading the list of 14 validated Oxstress mRNAs. 

### 4.11. Statistics

The non-parametric Mann–Whitney U-test was used to determine the statistical significance in demographics and in the microarray experiment. Data were expressed as mean (DE). A value of *p* < 0.05 was considered as statistically significant. Statistical analysis was performed with SPSS 20.0 and Volcano Plot was performed with Expression Suite software v1.1 (Thermofisher, Waltham, MA, USA) for determining interesting genes using (log) fold changes in conjunction with t-statistics. Transcripts levels in the scatterplot were calculated using the delta threshold cycle (ΔCt) method based on the relative quantification of *CCR7* or *FOXO1* normalized to *β-ACTIN* after determining the first cycle of fluorescence detection (ΔCt), and calculating the difference of these threshold cycles.

## 5. Conclusions

In conclusion, here we have deepened the functional relationship among miRNAs and transcripts associated to oxidative stress, a crucial step in the onset and progression of vascular disease. We have compiled a list of 417 genes associated to the response to oxidative stress (Oxstress genes) that after an integrated miRNA/mRNA counter-expression analysis, a protein–protein-interaction (PPI) network analysis and an expression profiling in PBMCs from patients suffering severe coronary artery disease (CAD), a serious consequence of ATHp, allowed the identification of FOXO1 and CCR7 as blood markers downregulated in CAD. 

## Figures and Tables

**Figure 1 ijms-21-01943-f001:**
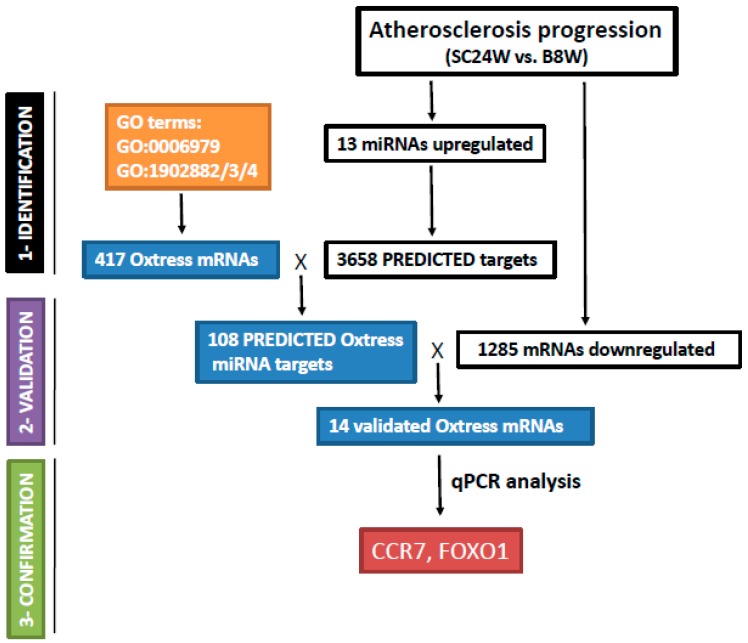
Diagram of the experimental approach used to identify putative miRNA-regulated Oxstress genes in atherosclerosis progression. Expression profiling of ATHp (SC24W vs. B8W, step 1, upper left box) had previously allowed the detection of 13 upregulated miRNAs, 1285 downregulated mRNAs, and 3,658 predicted miRNA targets [15,16]. A reverse GO term analysis of oxidative stress genes yielded 417 mRNAs for the primary GO:0006979 (response to oxidative stress) and its child terms GO:1902882/3/4 (see text for more details). This list was crossed with the list of 3658 predicted miRNA targets (see Materials and Methods) and the 108 entries common to both lists, likely representing Oxstress genes targeted by any of the 13 miRNAs, were saved. The list of 108 entries was crossed the list of 1,285 downregulated mRNAs to identify those actually downregulated during ATHp (step 2, middle left box). Expression of the 14 entries common to both lists (validated Oxstress mRNAs) was tested by qPCR to confirm their involvement in ATHp (step 3, lower left box). Steps 1 and 2 were made in mice or with mice data, step 3 was made with human material. The boxes shadowed in colors correspond to the work here performed, while clear boxes correspond to the work previously described [15,16].

**Figure 2 ijms-21-01943-f002:**
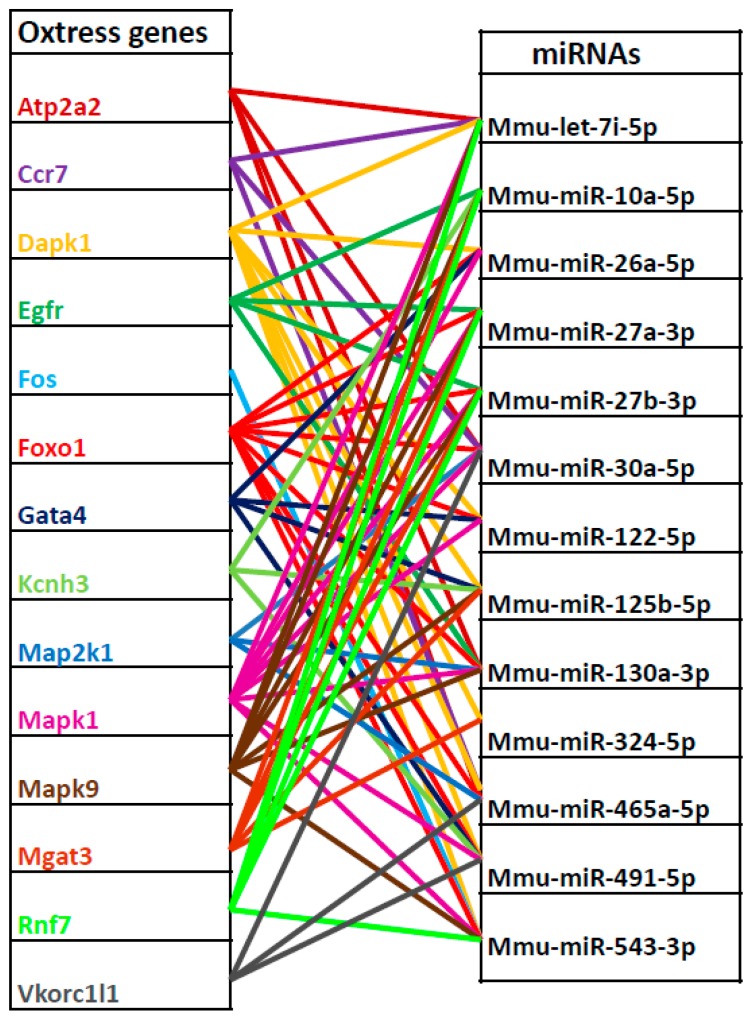
Predicted functional relationships among the 14 validated Oxstress genes and the 13 miRNAs upregulated during ATHp. Shown are, on the left the 14 validated Oxstress genes and on the right the 13 miRNAs upregulated during ATHp. Lines linking both columns indicate Oxstress transcripts that are predicted targets of the corresponding miRNAs. Data obtained from Table 2.

**Figure 3 ijms-21-01943-f003:**
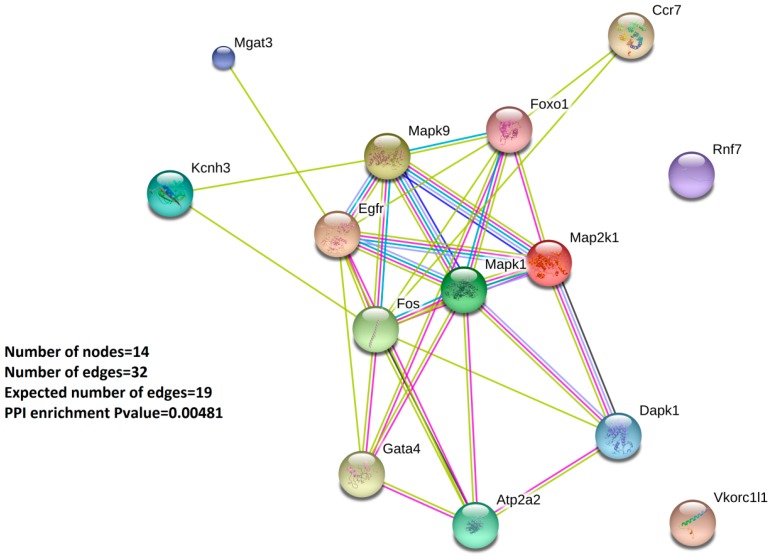
STRING Protein-Protein Interaction network of the 14 validated Oxstress mRNAs. Network nodes (colored circles) represent proteins, with a single node representing all the proteins produced by a single protein-coding gene, including splicing isoforms alternative polyadenylation forms, etc. Colored lines between the nodes (edges) indicate the different types of interaction evidenced by fusion of genes (red line), neighborhood of genes (green line), cooccurrence across species (blue line), experimental evidence (purple line), text mining of abstracts from literature (yellow line), databases (light blue line), co-expression in the same or others species (black line).

**Figure 4 ijms-21-01943-f004:**
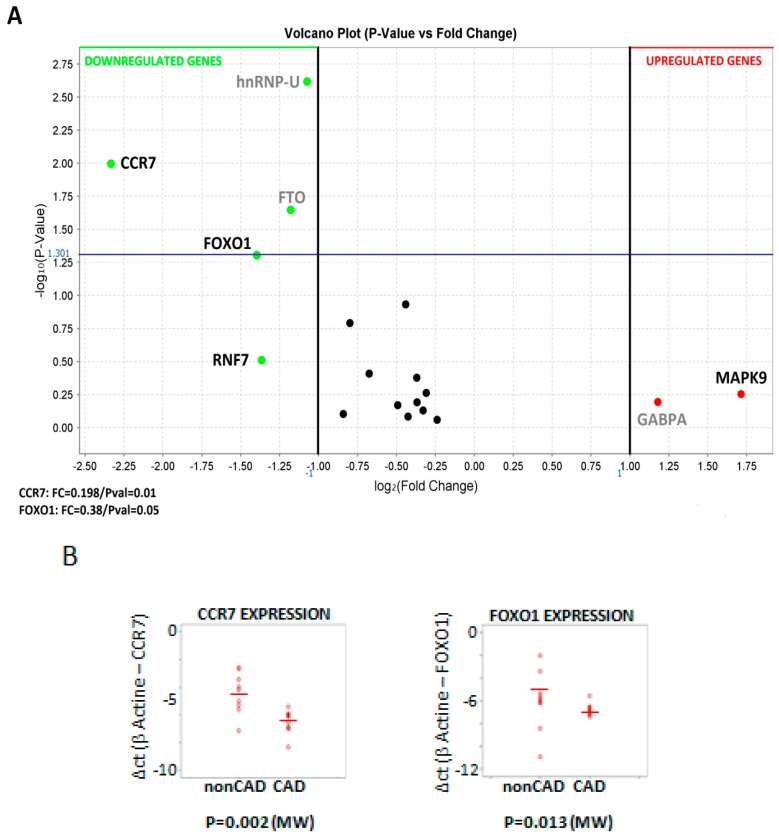
(**A**) Volcano plot (log_2_Fold Change vs. -log_10_(*p*-Value)) of the expression of the 14 validated Oxtress mRNAs in PBMCs of ATH patients. Colored dots identify upregulated genes (represented as red dots) or downregulated genes (green dots). The blue line indicates statistical significance. Gene Symbols in bold black indicate genes from this study while those in grey belonging to other study and will be published elsewhere. (**B**) Scatter Plot of *CCR7* and *FOXO1* expression in PBMNCs from patients with CAD and normal controls without CAD. The mean value and the *p* value are marked. *p* value was calculated with the nonparametric Mann-Whitney U test.

**Table 1 ijms-21-01943-t001:** Identity and expression of validated Oxstress transcripts. Shown are the Oxstress transcripts shown to be putative targets of the miRNAs whose expression was altered during ATHp or after treatment with the α-CD40 siRNA (as Fold Change in absolute values). Transcripts are identified by their unique GeneSymbol identifier and the specific isoform by its Seqname. Numbers between brackets refer to the groups of miRNAs of Appendix A. No downregulated transcripts predicted from upregulated miRNAs in the treatment group were identified.

Validated Oxstress Transcripts			
Atherosclerosis Progression (SC24W vs. B8W)			
Downregulated Transcripts (Predicted from Upregulated miRNAs [1])			
Gene Symbol	Seq Name	Fold Change (abs)	*p* Value
*Atp2a2*	NM_001110140	2.50	0011
*Ccr7*	NM_007719	13.63	0.019
*Dapk1*	NM_029653	3.90	0.002
*Egfr*	NM_007912	4.43	0.035
*Fos*	NM_010234	2.67	0.001
*Foxo1*	NM_019739	4.16	0.02
*Gata4*	NM_008092	3.63	0.001
*Kcnh3*	NM_010601	3.44	0.022
*Map2k1*	NM_008927	2.55	0.02
*Mapk1*	NM_011949	11.98	0.02
*Mapk9*	NM_001163671	2.85	0.005
*Mgat3*	NM_010795	27.02	0.005
*Rnf7*	NM_011279	33.12	0.02
*Vkorc1l1*	NM_001001327	3.95	0.03
**Upregulated Transcripts (Predicted from Downregulated miRNAs [2])**			
**Gene Symbol**	**Seq Name**	**Fold Change (abs)**	***p* Value**
*Pnkd*	NM_025580	4.88	0.01
**Treatment (α-siCD40/24W vs SC24W)**			
**Upregulated Transcripts (Predicted from Downregulated miRNAs [3])**			
**Gene Symbol**	**Seq Name**	**Fold Change (abs)**	***p* Value**
*App*	NM_001198826	2.32	0.03
*Cyth2*	NM_011181	2.97	0.0001
	NM_001112701	2.70	0.01
*Gm3286*	NM_001122678	2.15	0.010
*Macrod2*	NM_028387	2.41	0.02
	NM_001013802	2.43	0.03
*Sgk1*	NM_001161845	2.94	0.02
	NM_001161847	2.21	0.02
*Trmt2a*	NM_001080999	3.16	0.01
	NM_001195205	4.94	0.01

**Table 2 ijms-21-01943-t002:** Predicted miRNAs targeting each one of the 14 validated Oxstress transcripts. For each one of the validated Oxstress genes (identified by their gene symbol), shown are the predicted miRNAs targeting their 3’UTRs. Analysis was performed at the miRSystem browser (see Materials and Methods). In bold, miRNAs predicted to target both *Ccr7* and *Foxo1*.

MiRNAs Targetting validated Oxstress Transcripts	
Oxstress Transcripts	miRNAs
*Atp2a2*	mmu-let7i-5p; mmu-miR-130a-3p; mmu-miR-30a-5p; mmu-miR-465a-5p
*Ccr7*	mmu-let7i-5p; **mmu-miR-30a-5p**; **mmu-miR-465a-5p**
*Dapk1*	mmu-let7i-5p; mmu-miR-122-5p; mmu-miR-125b-5p; mmu-miR-26a-5p; mmu-miR-324-5p; mmu-miR-465a-5p; mmu-miR-491-5p; mmu-miR-543-3p
*Egfr*	mmu-miR-10a-5p; mmu-miR-130a-3p; mmu-miR-27a-3p; mmu-miR-27b-3p
*Fos*	mmu-miR-543-3p
*Foxo1*	mmu-miR-122-5p; mmu-miR-130a-3p; mmu-miR-26a-5p; mmu-miR-27a-3p; mmu-miR-27b-3p; **mmu-miR-30a-5p**; **mmu-miR-465a-5p**; mmu-miR-491-5p; mmu-miR-543-3p
*Gata4*	mmu-miR-122-5p; mmu-miR-125b-5p; mmu-miR-26a-5p; mmu-miR-491-5p;
*Kcnh3*	mmu-miR-10a-5p; mmu-miR-125b-5p; mmu-miR-491-5p;
*Map2k1*	mmu-miR-130a-3p; mmu-miR-30a-5p; mmu-miR-465a-5p
*Mapk1*	mmu-let7i-5p; mmu-miR-122-5p; mmu-miR-130a-3p; mmu-miR-26a-5p; mmu-miR-27a-3p; mmu-miR-27b-3p; mmu-miR-30a-5p; mmu-miR-491-5p; mmu-miR-543-3p
*Mapk9*	mmu-let7i-5p; mmu-miR-10a-5p; mmu-miR-125b-5p; mmu-miR-130a-3p; miR-27a-3p; mmu-miR-27b-3p; mmu-miR-543-3p
*Mgat3*	mmu-miR-10a-5p; mmu-miR-125b-5p; mmu-miR-27a-3p; mmu-miR-27b-3p; mmu-miR-324-5p
*Rnf7*	mmu-let7i-5p;mmu-miR-10a-5p;mmu-miR-27a-3p; mmu-miR-27b-3p; mmu-miR-543-3p
*Vkorc1l1*	mmu-miR-30a-5p; mmu-miR-465a-5p; mmu-miR-491-5p

**Table 3 ijms-21-01943-t003:** GO Enrichment analysis for Validated Oxstress mRNAs (*n* = 14). Shown are the first 20 Functional Categories at the “GO molecular function” section. Enrichment FDR shows the statistical significance (*p*-value cutoff [False Discovery Rate] = 0.05). Genes in list indicates the number of genes of the list that are also included in the “Total genes” of the “Functional Category”. Analysis was performed at the “ShinyGOv0.60 Gene Ontology Enrichment Analysis” server (see Materials and Methods).

GO Enrichment Analysis for Validated Oxstress mRNAs (*n* = 14)			
Enrichment (FDR)	Genes in List	Total Genes	Functional Category
3.9 × 10^−5^	3	26	Mitogen-activated protein kinase kinase kinase binding
7.3 × 10^−5^	3	40	Protein ser/threo/tyrosine kinase activity
5.6 × 10^−4^	5	633	Protein kinase activity
5.6 × 10^−4^	2	13	MAP kinasekinaseactivity
5.7 × 10^−4^	2	16	MAP kinase activity
5.7 × 10^−4^	5	693	Transcription factor binding
5.7 × 10^−4^	5	735	Phosphotransferase activity, alcohol group as acceptor
5.7 × 10^−4^	5	722	Protein kinase binding
7.7 × 10^−4^	5	803	Kinase binding
8.6 × 10^−4^	5	840	Kinase activity
1.0 × 10^−3^	4	456	Protein serine/threonine kinase activity
1.1 × 10^−3^	6	1544	ATP binding
1.1 x10^−3^	7	2346	Transferase activity
1.1 × 10^−3^	5	983	Transferase activity, transferring phosphorus-containing groups
1.1 × 10^−3^	7	2327	Enzyme binding
1.1 × 10^−3^	3	206	Phosphatase binding
1.1 × 10^−3^	6	1621	Adenyl nucleotide binding
1.1 × 10^−3^	6	1609	Adenyl ribonucleotide binding
1.1 × 10^−3^	7	2243	Catalytic activity, acting on a protein
1.8 × 10^−3^	6	1798	Drug binding

**Table 4 ijms-21-01943-t004:** Demographics and Clinical characteristics of patients. Abbreviations: CAD = Coronary artery disease, NonCAD = patients without CAD, GFR = Glomerular Filtration Rate, calculated by CKD-EPI method, SBP = Systolic Blood Pressure, DBP = Diastolic Blood Pressure, BP = Blood Pressure, ACE = Angiotensin-converting-enzyme, ARB = Angiotensin-Receptor Blockers, CCB = Calcium Channel Blocker. Diabetes was considered if patients were taking antidiabetic treatment. *p* value was calculated with the nonparametric Mann-Whitney U test. Standard deviation shown in parenthesis for the cholesterol, SBP and DBP values.

Demographics and Clinical Characteristics of Patients
	CAD (*n* = 10)	nonCAD (*n* = 12)	*p*
Gender (Female/Male)	5/5	5/7	ns
Age (years)	70 (11)	67 (12)	ns
GFR (ml/min)	25 (13)	64 (30)	0.002
Diabetes (yes/no)	5/5	3/9	ns
Cholesterol (mg/dL)	173 (65)	197 (41)	ns
SBP (mmHg)	127 (17)	125 (18)	ns
DBP (mmHg)	7 (11)	73 (14)	ns
**DRUGS**
Statins (yes/no)	9/1	7/5	
BP treatment (mean number)	3.1 (0.5)	2.6 (1.3)	
ACE inhibitors (yes)	3	8	
ARB (yes)	3	2	
Β-blockers (yes)	6	5	
CCB (yes)	8	1	
Diuretics (yes)	5	10	
Anti-platelets therapy (yes)	7	1

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
