# Peer review of "Integrated miRNA/mRNA Counter-Expression Analysis Highlights Oxidative Stress-Related Genes *CCR7* and *FOXO1* as Blood Markers of Coronary Arterial Disease"

_ijms, 2020, doi:10.3390/ijms21061943_

Round 1

Reviewer 1 Report

No further comments.

Reviewer 2 Report

The authors have improved the manuscript with data from patients to demonstrate the feasibility from bioinformatic analyses. 

This manuscript is a resubmission of an earlier submission. The following is a list of the peer review reports and author responses from that submission.

Round 1

Reviewer 1 Report

In the present manuscript, the authors used bioinformatics approach to identify oxidative stress-related genes CCR7 and FOXO1 as blood markers of coronary arterial disease. However, some questions need to be addressed.

The 13 miRNAs and 1,285 mRNAs were identified from ApoE-/- murine model mice aortas after systemic treatment with and anti-CD40 specific siRNA. In this study, the authors used a bioinformatics approach to deepen in the miRNA-dependent regulation of gene expression in ATHp; however, the results were experimentally validated in human PBMCs. What is the correlation between the miRNA and mRNA expression profiles in mice aortas and human PMBCs? Was the expression profile in mice aortas similar with mice peripheral blood mononuclear cells? In addition, the 13 miRNAs and 1,285 mRNAs were identified after anti-CD40 specific siRNA treatment. Did the patients with coronary artery disease receive any treatments? Why was the list from gene ontology crossed with the list from anti-CD40 siRNA treated mice aortas of mice? The figure 3B summarized the results in a volcano plot. It is not easy to observe the distribution of each gene expression. Since the samples were collected from more than 10 people, the scatter plot of each gene should be added. At least, the expression of CCR7 and FOXO1 should be shown. MiRNAs miR-30a-5p and miR-465a-400-5p was the predicted upstream of CCR7 and FOXO1. The expression level of miR-30a-5p and miR-465a-400-5p should be determined in human samples. The resolution of figure 3A and 3B should be improved.

Author Response

1.- What is the correlation between the miRNA and mRNA expression profiles in mice aortas and human PMBCs? Was the expression profile in mice aortas similar with mice peripheral blood mononuclear cells?

response:It is difficult to compare among the expression profiles of mice and human since we used a mutant model mouse (ApoE-deficient), fed with a high fat diet and submitted to a gene therapy with synthetic nucleotides, either anti-CD40 or the scrambled sequence control. We used human PBMCs from CAD patients and controls to validate mice data. We have added a paragraph to the Results section (lines 113-114, 231-232) to highlight this point.

2.- In addition, the 13 miRNAs and 1,285 mRNAs were identified after anti-CD40 specific siRNA treatment. Did the patients with coronary artery disease receive any treatments?

response:Yes, these are described in the new Table 4. Basically, these were lipid lowering drugs and/or blood pressure lowering drugs, ACE inhibitors, Angiotensin II Receptor Blockers (ARB), Calcium Channel Blocker (CCBs), β-blockers, diuretics or antiplatelet therapy.

3.- Why was the list from gene ontology crossed with the list from anti-CD40 siRNA treated mice aortas of mice?

response:Just to complete the data presented in the manuscript. If requested this data can be deleted

4.- The figure 3B summarized the results in a volcano plot. It is not easy to observe the distribution of each gene expression.

response:Volcano plots are a very intuitive and widely used representation of gene expression of large datasets, but the reviewer is right in that it is difficult to determine the actual values for the "Fold Change" and the "P-Value" before their log-transformations. These have been added as a small insert in Figure 4A.

5.- Since the samples were collected from more than 10 people, the scatter plot of each gene should be added. At least, the expression of CCR7 and FOXO1 should be shown.

response:This has been added as Figure 4B.

6.- MiRNAs miR-30a-5p and miR-465a-5p was the predicted upstream of CCR7 and FOXO1. The expression level of miR-30a-5p and miR-465a-5p should be determined in human samples.

response:We profiled expression of miR-30a in the human PBMCs but its expression was too low to be detected. Strikingly, mmu-miR-465a do not have a human homolog. This has been added to the results section and discussed in the Discussion section (lines 246-250 and 334-345).

7.- The resolution of figure 3A and 3B should be improved.

response:Figure 3 has been split into two (Figure 3 and 4A) and new Figures 3 and 4A have been provided at a higher resolution.

Reviewer 2 Report

The manuscript by Miguel Hueso et al. used Gene Ontology analysis to compile a list of genes associated with oxidative stress (Oxstress).  They then predicted the miRNAs targeted at these genes.  They then used protein-protein interaction network analysis identified FOXO1 and CCR7as blood markers downregulated in CAD.  The strategy was interesting, which yielded some potential candidate genes.

However, this study did not have any biological results to confirm the potential of FOXO1 and CCR7 in CAD.  It would be necessary for the authors to obtain some biological samples to demonstrate the power of  the bioinformatics analysis.

Minor comments:

The abbreviation should be Oxstress.  Please correct them.

Author Response

1.-However, this study did not have any biological results to confirm the potential of FOXO1 and CCR7 in CAD. It would be necessary for the authors to obtain some biological samples to demonstrate the power of the bioinformatics analysis. The last part of the Results section "

response:Experimental validation highlights the downregulation of CCR7 and FOXO1 in PBMCs from human CAD patients" was made in human biological samples of peripheral blood mononuclear cells from CAD patients and nonCAD controls. To highlight this, we have rewritten part of the section (lines 231-233).

2.- The abbreviation should be Oxstress. Please correct them.

response:Done